# Assessing the Efficacy of Mechanical Thrombectomy in Patients with an NIHSS < 6 Presenting with Proximal Middle Cerebral Artery Vessel Occlusion as Compared to Best Medical Management

**DOI:** 10.3390/brainsci13020214

**Published:** 2023-01-27

**Authors:** Vivek Srikar Yedavalli, Omar Hamam, Julie Gudenkauf, Richard Wang, Rafael Llinas, Elisabeth Breese Marsh, Justin Caplan, Kambiz Nael, Victor Urrutia

**Affiliations:** 1Russell H. Morgan Department of Radiology and Radiological Sciences, Johns Hopkins Hospital, 600 N Wolfe Street, Baltimore, MD 21287, USA; 2Department of Radiology and Radiological Sciences, Johns Hopkins Medicine, Baltimore, MD 21205, USA; 3Department Radiology and Radiological Sciences, Johns Hopkins Hospital, Baltimore, MD 21287, USA; 4Department of Radiology, University of California, Los Angeles, CA 90095, USA

**Keywords:** acute ischemic stroke, large vessel occlusion, mechanical thrombectomy, minor stroke, stroke care

## Abstract

Background and Purpose: Minor acute ischemic stroke (AIS) patients—defined by an NIHSS score < 6—presenting with proximal middle cerebral artery large vessel occlusions (MCA-LVO) is a subgroup for which treatment is still debated. Although these patients present with minor symptoms initially, studies have shown that several patients afflicted with MCA-LVO in this subgroup experience cognitive and functional decline. Although mechanical thrombectomy (MT) is the standard of care for patients with an NIHSS score of 6 or higher, treatment in the minor stroke subgroup is still being explored. The purpose of this preliminary study is to report our center’s experience in evaluating the potential benefit of mechanical thrombectomy (MT) in minor stroke patients when compared to medical management (MM). Methods: We performed a retrospective study with two comprehensive stroke centers within our hospital enterprise of consecutive patients presenting with minor AIS secondary to MCA-LVO (defined as M1 or proximal M2 segments of MCA). We subsequently evaluated patients who received MT versus those who received MM. Results: Between January 2017 and July 2021, we identified 46 AIS patients (11 treated with MT and 35 treated with MM) who presented with an NIHSS score < 6 secondary to MCA-LVO (47.8% 22/46 female, mean age 62.3 years, range 49–75 years). MT was associated with a significantly lower mRS at 90 days (median: 1.0 [IQR 0.0–2.0] versus 3.0 [IQR 1.0–4.0], *p* = <0.001), a favorable NIHSS shift (−4.0 [IQR −10.0–−2.0] versus 0.0 [IQR −2.0–1.0], *p* = 0.002), favorable NIHSS shift dichotomization (5/11, 45.5% versus 3/35, 8.6%, *p* = 0.003) and favorable mRS dichotomization (7/11, 63.6% versus 14/35, 40.0%, *p* = 0.024). Conclusions: In our center’s preliminary experience, for AIS patients presenting with an NIHSS score < 6 secondary to MCA-LVO, MT may be associated with improved clinical outcomes when compared to MM only.

## 1. Introduction

Stroke is one of the leading causes of disability in the US [1]. Patients with a low NIHSS score (<6) or minor stroke—a subset excluded from the most recent landmark trials [2]—are a subgroup for which treatment is still debated [3]. Current American Heart Association/American Stroke Association (AHA/ASA) guidelines consider treatment with IV thrombolysis (IV tPA) when NIHSS < 6 to be a reasonable approach [1].

Although an acute ischemic stroke (AIS) with NIHSS < 6 is considered by many to be minor and non-disabling, it can result in significant deficits in cognition and functional outcomes [4]. In addition, approximately 25% of these patients present with large vessel occlusion (LVO), the majority of which involve the middle cerebral artery [2]. Endovascular therapy, such as mechanical thrombectomy (MT), while the standard of care for patients with AIS secondary to MCA-LVO, is still controversial in patients presenting initially with minor stroke symptoms. Recent studies have demonstrated mixed results [5], with some showing potential benefits [6,7,8]. Therefore, the true benefit of MT has yet to be determined in larger scale studies.

The objective of this paper is to report our center’s preliminary experience in assessing whether MT affects clinical outcomes in AIS patients with secondary to middle cerebral artery LVO (MCA-LVO) presenting with an NIHSS score < 6 when compared to medical management (MM) alone.

## 2. Methods

### 2.1. Population

The data for the study population were collected using the Johns Hopkins Hospital and Johns Hopkins Bayview Medical Center Comprehensive Stroke Center Databases. This study was approved by the Johns Hopkins School of Medicine institutional review board (JHU-IRB00269637). In this retrospective analysis, we identified consecutive patients presenting with AIS using CT angiography (CTA), MR angiography (MRA), or digital subtraction angiography (DSA), and confirmed MCA-LVO and presenting with an NIHSS score < 6 within 24 h of symptom onset from 1 January 2017 to 7 January 2021. MCA-LVO was defined as the M1 or proximal M2 segments of the MCA [9,10]. Medical management was defined as any therapy other than MT and inclusive of intravenous thrombolysis (IV tPA).

### 2.2. Data Collection

Baseline and clinical data collected for each patient included demographics, risk factors for AIS (heart disease, hypertension, hyperlipidemia, diabetes mellitus, atrial fibrillation, prior stroke, chronic kidney disease, malignancy, smoking status, alcohol status, and BMI), admission NIHSS score, antiplatelet therapy, site of occlusion, laterality of occlusion, ASPECTS score, modified Rankin score at 90 days, and discharge NIHSS score. For each patient we also calculated the NIHSS shift (the difference between discharge and admission NIHSS scores). A favorable NIHSS shift was defined as an absolute decrease of four or greater [11].

The ASPECTS score and both the presence and site of MCA-LVO on CTA/MRA/DSA were assessed by a board certified neuroradiologist (VSY, 6 years of experience). Proximal M2 occlusion was defined as the vertical segment within the mesial margin of the sylvian fissure, as defined by Sarraj et al. [12]. Treatment type, if any, including IV tPA, MT, or both were noted. Interventional parameters, such as time to groin puncture, puncture to recanalization, modified thrombolysis in cerebral infarction (mTICI) score, MT method, and number of passes, were collected. Post reperfusion therapy hemorrhagic transformation (HT) and death were noted as safety endpoints when applicable. Patients were then subgrouped to MT versus MM for comparative analysis. Of note, patients with rescue thrombectomy performed 24 h after symptom onset were included in the MM subgroup.

### 2.3. Study Outcomes

The main outcome measurement of the study was the 90-day modified Rankin score (mRS). The outcome dichotomization between (mRS ≤ 1) versus (mRS 2–6); favorable (mRS ≤ 2) and unfavorable (mRS > 2), the NIHSS shift (discharge NIHSS score—admission NIHSS score), and a favorable NIHSS shift dichotomization (an NIHSS score shift decrease of four or greater) were used as secondary outcomes.

### 2.4. Statistical Analysis

The collected data were coded, tabulated, and statistically analyzed using IBM Statistical Package for Social Sciences (SPSS) statistics software version 28.0, IBM Corp., Chicago, IL, USA, 2021. The NIHSS shift was the difference between discharge and admission NIHSS scores; deceased cases were considered as non-improved. Quantitative data were tested for normality using the Shapiro–Wilk test. If normally distributed, data were described using mean ± SD (standard deviation) as well as minimum and maximum of the range. Non-normally distributed descriptions were described using the median (1st−3rd interquartiles) as well as the range minimum and maximum, and then compared using the Mann–Whitney test; *p* values of <0.05 were considered statistically significant.

## 3. Results

Among 111 consecutive patients presenting with NIHSS scores < 6 who were initially reviewed, 46 were enrolled with confirmed diagnosis of MCA-LVO; 35 patients were treated with MM, and 11 patients underwent MT. Two patients (2/35, 5.7%) within the M1 subgroup underwent rescue MT at least 24 h after symptom onset and were therefore included in the MM subgroup. The LVO breakdown in the MT group was: M1 (*n* = 8, 72.7%) and proximal M2 (*n* = 3, 27.3%), and in the MM group: M1 (*n* = 31, 88.6%) and proximal M2 (*n* = 4, 11.4%). Within the MT group, 6/11 (54.5%) received IV tPA, whereas 5/35 (14.2%) received IV tPA in the MM group.

The breakdown of the type of MT device used in the MT subgroup was as follows: suction device only (4/11, 36.4%), retrievable stent (4/11, 36.4%), angioplasty (2/11, 18.2%), and aspiration (1/11, 9.1%).

Demographics, risk factors and baseline characteristics are summarized in Table 1, based on treatment subgroups. The MT group presented with higher admission NIHSS scores (median: 3.0 [IQR 2.0–4.0] versus 2.0 [0.0–3.0], *p* = 0.033). The M1 subgroup analysis revealed a higher percentage of males in the MT subgroup (7/8, 87.5% versus 13/35, 41.9% in the MM subgroup) and a higher percentage of females in the MM subgroup (18/35, 58.1% versus 1/8, 12.5% in the MT subgroup) (*p* = 0.044).

Time parameters, IV tPA administration, type of thrombectomy device used, and number of passes are summarized in Table 2. The MT subgroup presented with shorter time from last known well to door in minutes (median: 17.0 [IQR 11.0–37.0] versus 43.0 [17.0–177.0], *p* = 0.041) and were administered IV tPA at a higher percentage (54.5% versus 14.3% in the MM subgroup, *p* = 0.013). Similarly, in our M1 subgroup analysis, the MT subgroup revealed a higher percentage of IV tPA administration compared to the MM subgroup (4/8, 50% versus 3/31, 9.7%, *p* = 0.022).

Outcome measures based on treatment with M1 subgroup analysis are summarized in Table 3. MT was associated with a significantly lower mRS at 90 days (1.0 [IQR 0.0–2.0] versus 3.0 [1.0–4.0], *p* = 0.012), a favorable NIHSS shift (−4.0 [IQR −10.0–2.0] versus 0.0 [−2.0–1.0], *p* = 0.002), favorable NIHSS shift dichotomization (5/11, 45.5% versus 3/35, 8.6%, *p* = 0.003) and favorable mRS dichotomization (7/11, 63.6% versus 14/35, 40.0%, *p* = 0.024). No significant difference was noted based on excellent outcome dichotomization. One case of HT was noted in the MT group (1/11, 9.1%). This patient received IV tPA. In the MM subgroup, 6/35 (17.1%) cases of HT were noted with four patients receiving IV tPA (4/35, 11.4%). Two deaths were noted in the MM group, but none in the MT group. Both deaths in the MM group were due to HT.

M1 subgroup analysis revealed that MT was associated with a significantly lower mRS at 90 days (1.0 [IQR 0.5–2.0] versus 3.0 [1.0–4.0], *p* = 0.012), a favorable NIHSS shift (−4.5 [IQR −10.0–3.0] versus 0.0 [−1.0–1.0], *p* = < 0.001), favorable NIHSS shift dichotomization (5/8, 62.5% versus 3/31, 9.7%, *p* = 0.003), and favorable mRS dichotomization (7/8, 87.5% versus 14/31, 45.2%, *p* = 0.049).

## 4. Discussion

Our center’s preliminary experience regarding retrospective evaluation suggests that MT may be beneficial in patients presenting with minor stroke (an NIHSS score < 6) with MCA-LVO treated as compared to MM alone.

Management of MCA-LVO for patients with low NIHSS scores continues to be debated. Studies exploring the low NIHSS score LVO subgroup have grown significantly within the past five years, particularly with respect to assessing the efficacy of MT in this group [6,13,14,15,16]. Griesenauer et al. [17] demonstrated that patients treated with MT were associated with better functional outcomes at 90 days compared to those who were not. Heldner et al. showed that patients who did not receive reperfusion therapy (defined as IV tPA, MT, or both) were more likely to experience neurological deterioration at 3 months [18]. Additionally, prior studies have also shown favorable outcomes associated with reperfusion therapy compared to the best medical management subgroup [14,19]. However, more recently, a multicenter study by Volny et al. [5] showed similar proportions of excellent outcomes between the MT and MM groups. This has been corroborated by other multicenter studies similarly investigating MT versus MM [20,21].

Our results are consistent with recent studies showing favorable outcomes with LVOs (including predominantly MCA-LVOs) when treated with MT [14,17,18,22]. Our population differs from Haussen et al.’s population due to our exclusion of basilar artery and ACA occlusions, but includes a similar population to Da Ros et al.’s [22]. Our proximal MCA analysis demonstrated improvement with MT, similar to previous investigations [8,13,14,15,19,23]. Other prior investigations found no benefit associated with MT in these subgroup populations [21,24,25,26,27]; notably, many of these studies (that did not note a benefit with MT in this vessel subgroup) dichotomized based on excellent outcomes [21,24,25,26] instead of favorable outcomes. This lack of benefit of MT with respect to excellent outcomes is in contrast to a recent study by Alexandre et al. They reported a benefit of MT in this population with respect to excellent outcomes, but not with favorable outcome dichotomization [28]. Our study did not find a benefit when dichotomized based on excellent outcomes, although we found a higher percentage of excellent outcomes in our MT subgroup as compared to the MM subgroup (5/11, 62.5% versus 9/35, 29.0%), which is similar to prior studies, including Alexandre et al. Furthermore, the lack of statistical significance between favorable and excellent outcome dichotomization may be due to the relatively low NIHSS on presentation resulting in a ceiling effect, or the lack of sensitivity of the outcome metrics to detect other disabling deficits such as cognitive impairment.

Although prior studies favor MT with M1 occlusions, the benefit of MT in isolated proximal M2 occlusions within the minor stroke population is still being debated. For instance, recently, Alexandre et al. [29] and Dobrocky et al. [30] concluded that MM demonstrated similar results to MT in isolated acute M2 occlusions. Although our sample was primarily comprised of M1s, we also included proximal M2 occlusions, as this subset of occlusions has been included in prior trials [9,31,32]. Given the preliminary nature of our study, however, our sample size of M2s was small, precluding dedicated subgroup analysis. Treatment of M2 occlusions still lacks consensus and requires a larger trial for definitive assessment.

It is important to note the risks of MT; the most concerning risk is intracerebral hemorrhage (ICH). Given the altered risk/benefit ratio, the relative safety of MT remains debated in this low NIHSS score group. Sarraj et al. noted a higher rate of ICH (5.8%) in their MT group without a difference in outcomes [20]. Another study also found a similar percentage of ICH with MT [6]. In contrast, several studies also found MT to be safe and effective in both M1 [13,15,19] and M2 occlusions [6]. In our study, 1 out of 11 in the MT group (9.1%) and 6 out of 35 (17.1%) in the MM subgroup resulted in post treatment hemorrhage, which are both within the range of prior reports [33,34]. The higher percentages in both subgroups may be a function of the small sample size. Notably, our post treatment hemorrhagic transformation results did not translate to mortality, as no patients expired in the MT subgroup, whereas two (2/35, 5.7%) died in the MM subgroup. Nevertheless, the difference in mortality was not statistically significant. Our results are therefore concordant with prior reports concluding that MT may also be safe.

Our study had several limitations to acknowledge. First, our preliminary study had the intrinsic limitations associated with retrospective analysis, limiting generalizability. To mitigate the effect of this limitation, we included both comprehensive stroke centers within our larger hospital enterprise. Second, our overall sample size of 46 patients was smaller compared to previous studies, which limited the strength of the study; therefore, our results must be interpreted with caution. However, we chose comparably more stringent criteria of LVO, limiting the definition to proximal MCA only comprised of M1 and proximal M2 occlusions. Notably, our sample size of MCA-LVO was comparable with several prior investigations [6,15,19,22]. Our sample M1 subgroup, specifically, was larger than the same subgroup in several prior reports; however, our M2 subgroup was substantially smaller [6,15]. Third, a higher percentage of patients within the MT subgroup received IV tPA compared to the MM subgroup, which may have contributed to the improved outcomes in the MT subgroup. Notably, higher IV tPA administration in MT subgroups was similarly noted in prior studies, and reflects the heterogeneity of treatment in current practice [21,23,35,36,37]. Fourth, patients in the MT subgroup presented with higher admission NIHSS scores and shorter times from last known well to door, which may have introduced selection bias. Last, in the MT subgroup, the majority were right sided LVOs (76.9%, 10/13). Prior studies have shown that patients with right sided LVOs tend to have lower admission NIHSS scores than left sided LVOs accounting for the same spatial location and size of ischemic core [38,39]. If some of these occlusions were left sided, they may not have qualified for the study. Nevertheless, cohorts assessed in prior studies also had a right sided LVO predominance [20,40]. This difference in admission NIHSS scores assessment underscores some of its inherent limitations, especially in the minor stroke population. For instance, prior studies have noted that the NIHSS may not capture the entire range of deficits, most notably with respect to cognitive deficits, gait abnormalities, and motor deficits [41].

## 5. Conclusions

In our center’s preliminary experience, treatment with MT appears to result in improved outcomes in patients presenting with an NIHSS score < 6 and MCA-LVO when compared to MM. A randomized trial is warranted to definitively confirm these results.

## Figures and Tables

**Table 1 brainsci-13-00214-t001:** Baseline characteristics by treatment.

Variables			All Cases			M1		
			MT (*N* = 11)	Medical (*N* = 35)	*p*-Value	MT (*N* = 8)	Medical (*N* = 31)	*p*-Value
**Age (years)**			66.5 ± 10.2	61.0 ± 13.7	^ 0.232	68.0 ± 8.5	61.5 ± 14.3	^ 0.231
**BMI (kg/m^2^)**			28.9 ± 10.7	30.3 ± 7.2	^ 0.646	26.2 ± 4.5	30.7 ± 7.4	^ 0.140
**Sex**	Male		8 (72.7%)	16 (45.7%)	§ 0.171	7 (87.5%)	13 (41.9%)	**§ 0.044 ***
	Female		3 (27.3%)	19 (54.3%)		1 (12.5%)	18 (58.1%)	
**Race**	Black		4 (36.4%)	17 (48.6%)	§ 0.516	3 (37.5%)	16 (51.6%)	§ 0.364
	White		5 (45.5%)	16 (45.7%)		3 (37.5%)	13 (41.9%)	
	Others		2 (18.2%)	2 (5.7%)		2 (25.0%)	2 (6.5%)	
**Current smoking**			5 (45.5%)	17 (48.6%)	§ 0.999	3 (37.5%)	16 (51.6%)	§ 0.695
**Current alcohol**			1 (9.1%)	8 (22.9%)	§ 0.421	0 (0.0%)	6 (19.4%)	§ 0.313
**Hypertension**			9 (81.8%)	28 (80.0%)	§ 0.999	6 (75.0%)	26 (83.9%)	§ 0.999
**Diabetes mellitus**			3 (27.3%)	11 (31.4%)	§ 0.999	3 (37.5%)	11 (35.5%)	§ 0.999
**Hyperlipidemia**			5 (45.5%)	16 (45.7%)	§ 0.999	4 (50.0%)	15 (48.4%)	§ 0.999
**Heart disease**			5 (45.5%)	12 (34.3%)	§ 0.722	3 (37.5%)	11 (35.5%)	§ 0.999
**Atrial Fibrillation**			4 (36.4%)	4 (11.4%)	§ 0.079	3 (37.5%)	4 (12.9%)	§ 0.137
**Previous stroke**			3 (27.3%)	8 (22.9%)	§ 0.999	2 (25.0%)	8 (25.8%)	§ 0.999
**Malignancy**			1 (9.1%)	7 (20.0%)	§ 0.658	1 (12.5%)	5 (16.1%)	§ 0.999
**Antiplatelet therapy**			3 (27.3%)	11 (31.4%)	§ 0.999	2 (25.0%)	10 (32.3%)	§ 0.999
**Chronic kidney disease**			0 (0.0%)	7 (20.0%)	§ 0.171	0 (0.0%)	6 (19.4%)	§ 0.313
**TOAST**		Large artery atherosclerosis	3 (27.3%)	16 (47.1%)	§ 0.397	3 (37.5%)	16 (53.3%)	§ 0.633
		Cardioembolism	7 (63.6%)	5 (14.7%)		4 (50.0%)	4 (13.3%)	
		Small-vessel occlusion	0 (0.0%)	1 (2.9%)		0 (0.0%)	1 (3.3%)	
		Stroke of other determined etiology	0 (0.0%)	3 (8.8%)		0 (0.0%)	2 (6.7%)	
		Stroke of undetermined etiology	1 (9.1%)	9 (26.5%)		1 (12.5%)	7 (23.3%)	
**ASPECTS**			9.6 ± 1.2	9.4 ± 1.1	^ 0.536	9.5 ± 1.4	9.4 ± 1.1	^ 0.825
**Admission NIHSS**			3.0 (3.0–4.0)	2.0 (0.0–3.0)	**¤ 0.033 ***	3.0 (3.0–4.0)	2.0 (1.0–3.0)	¤ 0.093
**Laterality**	Left		2 (18.2%)	19 (54.3%)	§0.067	2 (25.0%)	16 (51.6%)	§ 0.395
	Right		9 (81.8%)	15 (42.9%)		6 (75.0%)	14 (45.2%)	
	Bilateral		0 (0.0%)	1 (2.9%)		0 (0.0%)	1 (3.2%)	

BMI: Body mass index. Data presented as Mean ± SD or number (percent). ¤ Fishers Exact test. § Chi square test. ^ ANOVA test. * Significant (<0.050).

**Table 2 brainsci-13-00214-t002:** Time parameters and intervention by subgroup.

Variables			All Cases			M1		
			MT (*N* = 11)	Medical (*N* = 35)	*p*-Value	MT (*N* = 8)	Medical (*N* = 31)	*p*-Value
**Age (years)**			66.5 ± 10.2	61.0 ± 13.7	^ 0.232	68.0 ± 8.5	61.5 ± 14.3	^ 0.231
**BMI (kg/m^2^)**			28.9 ± 10.7	30.3 ± 7.2	^ 0.646	26.2 ± 4.5	30.7 ± 7.4	^ 0.140
**Sex**	Male		8 (72.7%)	16 (45.7%)	§ 0.171	7 (87.5%)	13 (41.9%)	**§ 0.044 ***
	Female		3 (27.3%)	19 (54.3%)		1 (12.5%)	18 (58.1%)	
**Race**	Black		4 (36.4%)	17 (48.6%)	§ 0.516	3 (37.5%)	16 (51.6%)	§0.364
	White		5 (45.5%)	16 (45.7%)		3 (37.5%)	13 (41.9%)	
	Others		2 (18.2%)	2 (5.7%)		2 (25.0%)	2 (6.5%)	
**Current smoking**			5 (45.5%)	17 (48.6%)	§ 0.999	3 (37.5%)	16 (51.6%)	§ 0.695
**Current alcohol**			1 (9.1%)	8 (22.9%)	§ 0.421	0 (0.0%)	6 (19.4%)	§ 0.313
**Hypertension**			9 (81.8%)	28 (80.0%)	§ 0.999	6 (75.0%)	26 (83.9%)	§ 0.999
**Diabetes mellitus**			3 (27.3%)	11 (31.4%)	§ 0.999	3 (37.5%)	11 (35.5%)	§ 0.999
**Hyperlipidemia**			5 (45.5%)	16 (45.7%)	§ 0.999	4 (50.0%)	15 (48.4%)	§ 0.999
**Heart disease**			5 (45.5%)	12 (34.3%)	§0.722	3 (37.5%)	11 (35.5%)	§ 0.999
**Atrial Fibrillation**			4 (36.4%)	4 (11.4%)	§ 0.079	3 (37.5%)	4 (12.9%)	§ 0.137
**Previous stroke**			3 (27.3%)	8 (22.9%)	§ 0.999	2 (25.0%)	8 (25.8%)	§ 0.999
**Malignancy**			1 (9.1%)	7 (20.0%)	§ 0.658	1 (12.5%)	5 (16.1%)	§ 0.999
**Antiplatelet therapy**			3 (27.3%)	11 (31.4%)	§ 0.999	2 (25.0%)	10 (32.3%)	§ 0.999
**Chronic kidney disease**			0 (0.0%)	7 (20.0%)	§ 0.171	0 (0.0%)	6 (19.4%)	§ 0.313
**TOAST**		Large artery atherosclerosis	3 (27.3%)	16 (47.1%)	§ 0.397	3 (37.5%)	16 (53.3%)	§ 0.633
		Cardioembolism	7 (63.6%)	5 (14.7%)		4 (50.0%)	4 (13.3%)	
		Small-vessel occlusion	0 (0.0%)	1 (2.9%)		0 (0.0%)	1 (3.3%)	
		Stroke of other determined etiology	0 (0.0%)	3 (8.8%)		0 (0.0%)	2 (6.7%)	
		Stroke of undetermined etiology	1 (9.1%)	9 (26.5%)		1 (12.5%)	7 (23.3%)	
**ASPECTS**			9.6 ± 1.2	9.4 ± 1.1	^ 0.536	9.5 ± 1.4	9.4 ± 1.1	^ 0.825
**Admission NIHSS**			3.0 (3.0–4.0)	2.0 (0.0–3.0)	**¤ 0.033 ***	3.0 (3.0–4.0)	2.0 (1.0–3.0)	¤ 0.093
**Laterality**	Left		2 (18.2%)	19 (54.3%)	§ 0.067	2 (25.0%)	16 (51.6%)	§ 0.395
	Right		9 (81.8%)	15 (42.9%)		6 (75.0%)	14 (45.2%)	
	Bilateral		0 (0.0%)	1 (2.9%)		0 (0.0%)	1 (3.2%)	

NA: Not applicable. Unless otherwise noted, data presented as Mean ± SD or number (percent). § Fishers Exact test. ^ Independent *t*-test. ¤ Mann Whitney test. * Significant (<0.050).

**Table 3 brainsci-13-00214-t003:** Outcomes with M1 subgroup analysis.

Outcomes		All Cases			M1	
		MT (*N* = 11)	Medical (*N* = 35)	*p*-Value	MT (*N* = 8)	Medical (*N* = 31)
**Post treatment hemorrhagic transformation**		1 (9.1%)	6 (17.4%)	0.314	1 (12.5%)	5 (16.2%)
**Discharge NIHSS**		1.0 (0.0–2.0)	1.0 (0.0–2.0)	¤ 0.642	0.5 (0.0–2.0)	2.0 (1.0–3.0)
**NIHSS shift**		−4.0 (−10.0–−2.0)	0.0 (-2.0–1.0)	**¤ 0.002 ***	−4.5 (−10.0–−3.0)	0.0 (−1.0–1.0)
**NIHSS Shift**	Favorable	5 (62.5%)	3 (9.7%)	**# 0.003 ***	5 (62.5%)	3 (9.7%)
	Unfavorable	3 (37.5%)	28 (90.3%)		3 (37.5%)	28 (90.3%)
**90 d mRS**		1.0 (0.0–2.0)	3.0 (1.0–4.0)	**¤ 0.012 ***	1.0 (0.5–2.0)	3.0 (1.0–4.0)
**Functional based on favorable 90 d MRS**	Good (0–2)	7 (87.5%)	14 (45.2%)	**# 0.024 ***	7 (87.5%)	14 (45.2%)
	Poor (3–6)	1 (12.5%)	17 (54.8%)		1 (12.5%)	17 (54.8%)
**Functional based on excellent 90 d MRS**	Excellent (0–1)	5 (62.5%)	9 (29.0%)	§ 0.153	5 (62.5%)	9 (29.0%)
	Poor (2–6)	3 (37.5%)	22 (71.0%)		3 (37.5%)	22 (71.0%)
**Mortality**		0 (0.0%)	2 (6.5%)	§ 0.999	0 (0.0%)	2 (6.5%)

NA: Not applicable. Unless otherwise noted, data presented as Mean ± SD or number (percent). § Fishers Exact test. # Chi square test. ^ Independent *t*-test. ¤ Mann Whitney test. * Significant (<0.050).

## Data Availability

Data can be provided upon request.

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
