# Peer review of "Assessing the Efficacy of Mechanical Thrombectomy in Patients with an NIHSS < 6 Presenting with Proximal Middle Cerebral Artery Vessel Occlusion as Compared to Best Medical Management"

_brainsci, 2023, doi:10.3390/brainsci13020214_

Round 1
Reviewer 1 Report
This is a good retrospective analysis looking for benefit of mechanical thrombectomy compared with medical management in patients with MCA-LVO but with NIHSS <6. The authors have presented a solid and wellorganised and analyzed data to support their conclusion, but a few questions remain:
1. As authors have themselves cited, a higher fraction of the patients in MT group received IV Alteplase compared with MM group. It is very much possible that thrombolysis may have resulted in better outcomes in MT group, and if a similar fraction had received IV Alteplase in the MM group, their outcomes may have been better as well. This fact/confounder should at least be acknowledged in discussion.
2. If this information is available, please mention if the hemorrhagic transformations in either group happened in the patients who received IV alteplase.
The following are a few minor comments which if addressed will make the manuscript technically correct:
1. By definition, the minor stroke is the term used for strokes with NIHSS 1-4. The authors' use of the term/definition for stroked <6 may be confusing for the readers. Please consider changing it. (Lines 14, 38, 45)
2. Line 97: Please consider writing a consistent definition of NIHSS shift (difference between NIHSS at discharge and admission) and not vice versa.
3. Lines 24 and 67: Please consider replacing the word 'and' with 'or' in definition of MCA-LVO.
4. Line 22: Please consider adding (MM) for medical management at this point before starting to use the abbreviation later in the manuscript.
5. Line 82: Please consider adding the word 'score' after m(TICI)
Author Response
REVIEWER 1:
This is a good retrospective analysis looking for benefit of mechanical thrombectomy compared with medical management in patients with MCA-LVO but with NIHSS <6. The authors have presented a solid and wellorganised and analyzed data to support their conclusion, but a few questions remain:
- As authors have themselves cited, a higher fraction of the patients in MT group received IV Alteplase compared with MM group. It is very much possible that thrombolysis may have resulted in better outcomes in MT group, and if a similar fraction had received IV Alteplase in the MM group, their outcomes may have been better as well. This fact/confounder should at least be acknowledged in discussion.
We thank the reviewer for this critique. Per the reviewer’s suggestion, we have included this within the limitations section more clearly. Please see lines 182-184.
- If this information is available, please mention if the hemorrhagic transformations in either group happened in the patients who received IV alteplase.
The single case of HT in the MT group received IV tPA while four out of the six cases of HT in the MM group received IV tPA. Please see lines 121-123.
The following are a few minor comments which if addressed will make the manuscript technically correct:
- By definition, the minor stroke is the term used for strokes with NIHSS 1-4. The authors' use of the term/definition for stroked <6 may be confusing for the readers. Please consider changing it. (Lines 14, 38, 45)
Thank you for the critique. The majority of the most recent studies have used the cutoff of admission NIHSS < 6. Please see references Sarraj et al, Nagel et al, Haussen et al, Da Ros et al, Messer et al, Goyal et al as examples.
- Line 97: Please consider writing a consistent definition of NIHSS shift (difference between NIHSS at discharge and admission) and not vice versa.
This has been modified per the reviewer’s suggestion. Please see line 86.
- Lines 24 and 67: Please consider replacing the word 'and' with 'or' in definition of MCA-LVO.
This has been modified per the reviewer’s suggestion. Please see lines 14 and 55.
- Line 22: Please consider adding (MM) for medical management at this point before starting to use the abbreviation later in the manuscript.
This has been modified per the reviewer’s suggestion. Please see line 12.
- Line 82: Please consider adding the word 'score' after m(TICI)
This has been modified per the reviewer’s suggestion. Please see line 71.
Reviewer 2 Report
Interesting and well written study Assessment the efficacy of mechanical thrombectomy in patients with an NIHSS < 6 presenting with proximal middle cerebral artery vessel occlusion as compared to best medical management.
The idea of the study is valuable, since it has great importance in common clinical practice.
In our opinion there are just some minor aspects to be improved before publication:
- Minor English correction needed.
- Materials and methods and results: could you please better clarify which are the involved segment occluded? and can you please specify the endovascular technique which has been used?
- Do you usually treat the patient immediately or do you wait for worsening before indication to thrombectomy?
- In which percentage of the treated patients thrombolysis was given?
- did you considere just LVO occlusions (such as ICA or M1), or did you consider even M2 occlusion?
Discussion: there are some other recently published experience regarding LVO occlusion in minor stroke (eg: Alexandre AM et al. Mechanical thrombectomy in acute ischemic stroke due to large vessel occlusion in the anterior circulation and low baseline National Institute of Health Stroke Scale score: a multicenter retrospective matched analysis. Neurol Sci. 2022 May;43(5):3105-3112. doi: 10.1007/s10072-021-05771-5. Epub 2021 Nov 29. PMID: 34843020), can you please comment your results compared to those ones?.
- Another important point is the occlusion of M2 segment of MCA, which in some cases results is major stroke (even if the NIHSS score remains low), in this case the indication to thrombectomy is much more debated and results are sometimes in favor of best medical therapy, including thrombolysis (eg: Alexandre AM et al. Mechanical thrombectomy in minor stroke due to isolated M2 occlusion: a multicenter retrospective matched analysis. J Neurointerv Surg. 2022 Oct 12:jnis-2022-019557. doi: 10.1136/jnis-2022-019557. Epub ahead of print. PMID: 36223995.), could you please add some lines in discussion regarding this point?
- Finally can you please comment results in anterior circulation LVO occlusion compared to posterior circulation LVO occlusion, always presenting with low NIHSS score (see for example: Alexandre AM et al. Posterior Circulation Endovascular Thrombectomy for Large Vessels Occlusion in Patients Presenting with NIHSS Score ≤ 10. Life (Basel). 2021 Dec 17;11(12):1423. doi: 10.3390/life11121423. PMID: 34947955; PMCID: PMC8703711.)
- Conclusion: could you please better underline which are the main findings of your study?
Author Response
REVIEWER 2:
Interesting and well written study Assessment the efficacy of mechanical thrombectomy in patients with an NIHSS < 6 presenting with proximal middle cerebral artery vessel occlusion as compared to best medical management.
The idea of the study is valuable, since it has great importance in common clinical practice.
In our opinion there are just some minor aspects to be improved before publication:
- Minor English correction needed.
- Materials and methods and results: could you please better clarify which are the involved segment occluded? and can you please specify the endovascular technique which has been used?
We included M1 and proximal M2 occlusions only. Please see the LVO breakdown within the results section (Please see lines 96-99). We also defined the occlusions within the population section of the methods (Please see lines 55-56).
Per the reviewer’s suggestion, we also added a paragraph with respect to the type of treatment technique (please see lines 101-102).
- Do you usually treat the patient immediately or do you wait for worsening before indication to thrombectomy?
The patients were treated immediately on consensus evaluation by the stroke neurologist, neurointerventionalist, and neuroradiologist. Two patients underwent rescue thrombectomy more than 24 hours after symptom onset and were included in the MM group for that reason.
- In which percentage of the treated patients thrombolysis was given?
In the MT group, 6/11 received IV tPA, while 5/35 received IV tPA in the MM group. Please see table 2 for more details.
- did you considere just LVO occlusions (such as ICA or M1), or did you consider even M2 occlusion?
We only included M1 and proximal M2 occlusions. Please see the population section of the methods for additional detail.
Discussion: there are some other recently published experience regarding LVO occlusion in minor stroke (eg: Alexandre AM et al. Mechanical thrombectomy in acute ischemic stroke due to large vessel occlusion in the anterior circulation and low baseline National Institute of Health Stroke Scale score: a multicenter retrospective matched analysis. Neurol Sci. 2022 May;43(5):3105-3112. doi: 10.1007/s10072-021-05771-5. Epub 2021 Nov 29. PMID: 34843020), can you please comment your results compared to those ones?.
We found similar benefit with MT compared to MM as seen in the reference from Alexandre et al provided by the reviewer. We have modified it to include this reference with discussion in lines 147-152.
- Another important point is the occlusion of M2 segment of MCA, which in some cases results is major stroke (even if the NIHSS score remains low), in this case the indication to thrombectomy is much more debated and results are sometimes in favor of best medical therapy, including thrombolysis (eg: Alexandre AM et al. Mechanical thrombectomy in minor stroke due to isolated M2 occlusion: a multicenter retrospective matched analysis. J Neurointerv Surg. 2022 Oct 12:jnis-2022-019557. doi: 10.1136/jnis-2022-019557. Epub ahead of print. PMID: 36223995.), could you please add some lines in discussion regarding this point?
We thank the reviewers for these comments and per the reviewer’s recommendation, we have added a section in the discussion which also references the study above. Please see lines 156-162.
We included proximal M2s, which was a minority of the overall sample (7/46). With the small sample size of M2s, we did not perform a separate subgroup analysis and instead focused on an M1 subgroup analysis.
- Finally can you please comment results in anterior circulation LVO occlusion compared to posterior circulation LVO occlusion, always presenting with low NIHSS score (see for example: Alexandre AM et al. Posterior Circulation Endovascular Thrombectomy for Large Vessels Occlusion in Patients Presenting with NIHSS Score ≤ 10. Life (Basel). 2021 Dec 17;11(12):1423. doi: 10.3390/life11121423. PMID: 34947955; PMCID: PMC8703711.)
We appreciate the reviewers comments on this interesting comparison. We felt that a contrast to posterior circulation occlusions was beyond the scope of the preliminary study as we focused solely on anterior circulation and proximal MCA occlusions specifically. Incidentally, our group is in the process of performing a larger study building on these results where we plan on including pc-LVOs after achieving the requisite sample size. We believe that this reference will be especially pertinent and essential to that follow up study.
- Conclusion: could you please better underline which are the main findings of your study?
Our main findings are the MT appears beneficial for patients presenting with NIHSS < 6 and proximal MCA-LVO as compared to medical management alone. However, given the preliminary nature of this study, we suggest that larger sample sizes are necessary to corroborate our results.
Reviewer 3 Report
This is a retrospective study in a subset of patients with an M1 occlusions and with low NIHSS scores at admission. The authors find that patients seem to have a better outcome if treated endovascularly.
The results are not suprising but still important in the discussion on whether or not to treat patients with an LVO and low NIHSS scores.
In the disscussion the author could add a few sentences on the flaws of the NIHSS score. For example, somebody with afasia might have a low NIHHS score but still has a very disabling deficit. For somebody presenting with a LVO and an NIHSS of 5 it is important to realize that this is still a potentially very disabling deficit.
Minor edit: line 193: 2/32 is not 95.7%
Author Response
REVIEWER 3:
This is a retrospective study in a subset of patients with an M1 occlusions and with low NIHSS scores at admission. The authors find that patients seem to have a better outcome if treated endovascularly.
The results are not suprising but still important in the discussion on whether or not to treat patients with an LVO and low NIHSS scores.
In the disscussion the author could add a few sentences on the flaws of the NIHSS score. For example, somebody with afasia might have a low NIHHS score but still has a very disabling deficit. For somebody presenting with a LVO and an NIHSS of 5 it is important to realize that this is still a potentially very disabling deficit.
We thank the reviewer for these insights. We have expanded our limitations section to address the limitations of the NIHSS stroke scale. Please see lines 192-194.
Minor edit: line 193: 2/32 is not 95.7%.
Thank you, this has been modified accordingly. Please see line 171.